# Selective sorting of polymers with different terminal groups using metal-organic frameworks

Benjamin Le Ouay[1,2], Chikara Watanabe[3], Shuto Mochizuki[3], Masayoshi Takayanagi[2,4,5], Masataka Nagaoka[2,6], Takashi Kitao [1,2,7] & Takashi Uemura [1,2,7]

Separation of high-molecular-weight polymers differing just by one monomeric unit remains a challenging task. Here, we describe a protocol using metal-organic frameworks (MOFs) for the efficient separation and purification of mixtures of polymers that differ only by their terminal groups. In this process, polymer chains are inserted by threading one of their extremities through a series of MOF nanowindows. Selected termini can be adjusted by tuning the MOF structure, and the insertion methodology. Accordingly, MOFs with permanently opened pores allow for the complete separation of poly(ethylene glycol) (PEG) based on steric hindrance of the terminal groups. Excellent separation is achieved, even for high molecular weights (20 kDa). Furthermore, the dynamic character of a flexible MOF is used to separate PEG mixtures with very similar terminal moieties, such as OH, OMe, and OEt, as the slight difference of polarity in these groups significantly changes the pore opening kinetics.

[1] Department of Advanced Materials Science, Graduate School of Frontier Sciences, The University of Tokyo, 5-1-5 Kashiwanoha, Kashiwa, Chiba 277-8561, Japan. [2] CREST, Japan Science and Technology Agency (JST), 4-1-8 Honcho, Kawaguchi, Saitama 332-0012, Japan. [3] Department of Synthetic Chemistry and Biological Chemistry, Graduate School of Engineering, Kyoto University, Katsura, Nishikyo-ku, Kyoto 615-8510, Japan. [4] The Center for Data Science Education and Research, Shiga University, 1-1-1 Banba, Hikone, Shiga 522-8522, Japan. [5] RIKEN Center for Advanced Intelligence Project, 1-4-1 Nihonbashi, Chuo-ku, Tokyo 103-0027, Japan. [6] Department of Complex Systems Science, Graduate School of Informatics, Nagoya University, Furo-cho, Chikusa-ku, Nagoya 464-8601, Japan. [7] Department of Applied Chemistry, Graduate School of Engineering, The University of Tokyo, 7-3-1 Hongo, Bunkyo-ku, Tokyo 113-8656, Japan. Correspondence and requests for materials should be addressed to T.U. (email: t-uemura@k.u-tokyo.ac.jp)

Polymers are ubiquitous in modern technology, but remain challenging to purify, especially when chains present only marginal structural differences. As such, while separation based on molecular weight can easily be accomplished with common chromatographic techniques such as gel permeation chromatography (GPC)[1,2], the separation of polymers with identical chains but different terminal groups is an extremely difficult task. Indeed, as the length of the chains increases, the few different groups become diluted among numerous identical monomeric units, and their impact on the physical properties (such as solubility or size of the coils) becomes negligible, making the separation with usual techniques almost impossible. The high number of monomeric units susceptible to bringing nonspecific interaction also lowers considerably the efficacy of chromatography techniques based on preferential affinity. To circumvent this effect, some techniques, such as liquid chromatography in critical conditions, have been developed, allowing for a separation based on terminal groups[3–5]. Formation of supramolecular assemblies with polymers has also been used to differentiate identical chains with different termini[6,7]. However, these methods operate in a very narrow range of conditions that need to be carefully tuned, and are out of reach for most nonspecialized laboratories[8]. These drawbacks bring the need for a robust, systematic, and high-throughput method to separate polymers precisely.

Metal-organic frameworks (MOFs) are constituted by the assembly of metal nodes and polytopic organic linkers that form a porous crystalline framework[9–11]. Thus, the pore volume consists of repeated nanocages, connected to each other by a series of nanowindows, with a well-defined pore opening and shape. Owing to their high and accessible porosity, virtually infinite structural and functional tunability, and flexible behavior, MOFs have been extensively studied for the adsorption and separation of gases and small molecules[12–16]. However, to date, the separation of large molecules remains limited to specific compounds with a molecular weight around 1000 g mol$^{-1}$[17,18]. Moreover, no report has described the separation of polymers, although the direct insertion of polymers into MOFs is possible, but rare[19–26]. One highly challenging yet technologically relevant polymer separation is the sorting of poly(ethylene glycol) (PEG) according to various terminal groups. While simple in structure, PEG is one of the most widely used polymers, with applications including bioconjugates[27,28], drug-release systems[29], surface and nanoparticle coating[30,31], and solid electrolytes[32]. PEG owes its success to a unique combination of nontoxicity, high solubility both in water and in organic solvents, nonimmunogenicity, overall low protein binding, and ion complexation capability. Because PEG is often coupled to other structures by one terminus through covalent bonds and also because some groups can bring parasite reactions[33], controlling the nature of both termini of a chain is essential for most applications. However, because of the lack of available separation methods so far, it is very difficult to assess and improve the purity of PEG derivatives, bringing a source of irreproducibility to experimental data.

In this article, we describe the utilization of MOFs for the separation of polymers with identical repeating units and chain length, but different terminal groups. Several MOFs are used to separate functionalized PEGs according to their termini, where R$^1$–(OC$_2$H$_4$)$_n$–OR$^2$ is referred to as R$^1$–PEG–R$^2$. Regardless of the length of the chain, the insertion occurs by threading one of the chain extremities through a series of repeated nanowindows. The nature of the terminal groups thus determines the condition for a possible insertion, and only polymer chains with adequate moieties are fully introduced, while other chains remain totally excluded from the MOF. As such, even if the terminal groups constitute a negligible fraction of their mass, polymer chains of high molecular weight can be separated as efficiently as if they were small molecules. To show the generality of the technique and bring more insight into the potential of MOFs, we investigate both rigid and flexible frameworks for the separation of PEG. Notably, the selectivity for some terminal groups was successfully modulated by the size and shape of the MOF pores, and by their dynamic character. As such, rigid MOFs prove very efficient for the separation of PEG with different terminal groups on the basis of steric hindrance, and the selectivity can be modulated with the choice of ligands. In addition, PEGs with termini having the same size, but a different polarity, can be finely separated using the dynamic pore system of a flexible MOF.

## Results

**Sorting of PEG by rigid MOFs**. [Zn$_2$(1,4-naphthalenedicarboxylate)$_2$triethylenediamine]$_n$ (**1a**; pore size = 5.7 × 5.7 Å$^2$)[34] with regular nanochannels along the c-axis was employed as a rigid host MOF (Fig. 1a and Supplementary Fig. 1). We first introduced H–PEG–H (2 kDa) into the nanochannels of **1a** as follows. The MOF compound **1a** was immersed in a MeCN solution of PEG, followed by removal of the solvent in vacuo at room temperature. Further thermal annealing above the melting temperature of PEG drove the complete insertion of polymer chains into the nanochannels, providing the host–guest composite between **1a** and PEG while maintaining the crystallinity of the host framework (Supplementary Fig. 2). The mass fraction of PEG was optimized below the maximal guest capacity in **1a**, allowing for full inclusion of PEG into the **1a** crystals with the size and morphology maintained, as confirmed by scanning electron microscopy (SEM) and particle size distribution analysis (Supplementary Figs 3 and 4). **1a** including PEG exhibited significantly lower adsorption capacity for nitrogen because of the pore occupancy with PEG (Supplementary Fig. 5). Multinuclear 2-D nuclear magnetic resonance (NMR) is one of the most powerful methods to provide direct evidence of the encapsulation of polymer chains in pores[35]. Notably, the 2-D $^1$H–$^{13}$C heteronuclear correlated (HETCOR) NMR spectrum of the **1a** and PEG composite highlighted a cross-peak associated with the intermolecular host–guest interactions that occurred through dipole–dipole interactions at short distances of less than 5 Å (Fig. 1b). This is a clear indication that PEG was accommodated in the nanochannels of **1a**. Formation of the host–guest composites was analyzed further using differential scanning calorimetry (DSC) measurements. In general, the DSC of polymers confined in nanopores does not display any peaks corresponding to thermal transitions of polymers in the bulk state, such as melting and glass transition[20,35]. Actually, the DSC for **1a** with H–PEG–H showed a remarkable decrease in the melting enthalpy of PEG even before the annealing process, suggesting that even at room temperature, a part (ca. 30 wt%) of the PEG used was already included in the nanochannels via solvent removal. Subsequent thermal treatment above the melting point of PEG led to full insertion of the molten PEG into **1a**, resulting in the complete disappearance of the endothermic peak for the final **1a** and PEG composite (Fig. 1c and Supplementary Fig. 6). The inclusion of H–PEG–H in **1a** was enthalpically driven by the interaction between PEG and pore walls (Supplementary Fig. 7)[36]. The kinetics of this insertion was also determined using DSC (Supplementary Fig. 8).

We examined the introduction of PEG with other terminal groups into **1a**, revealing the different insertion behavior depending on the terminal groups. Note that PEG with small terminal groups, Me–PEG–Me, could be also introduced into the nanochannels of **1a**, as evidenced by a series of characterizations (Supplementary Figs 2, 3, and 9). With the same diameter of chain, Me–PEG–Me and H–PEG–H were found to present similar insertion kinetics (Supplementary Fig. 8). In contrast, PEG modified with large trityl (Tr) groups (size = 7.2 Å), Tr–PEG–Tr,

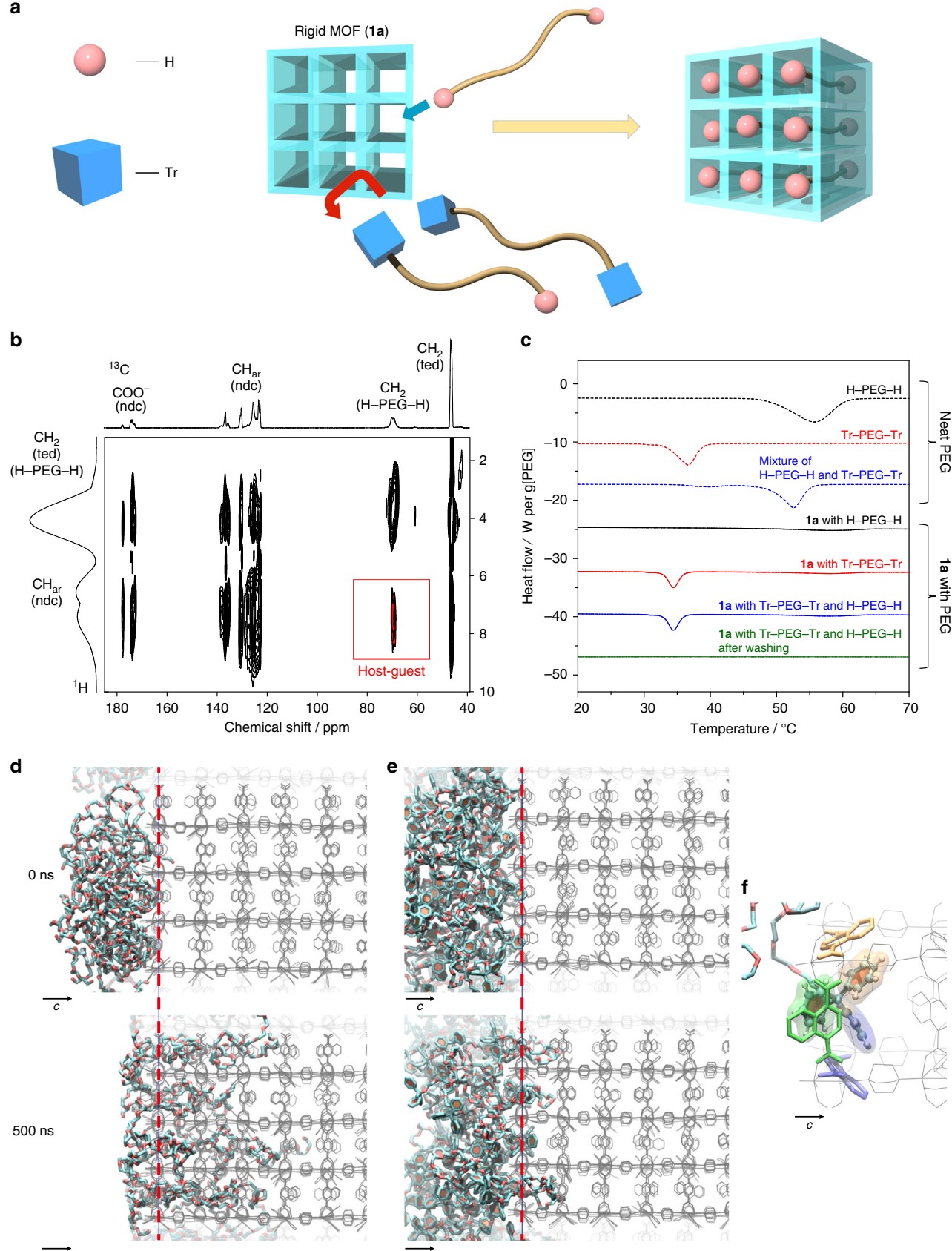

**Fig. 1** Use of **1a** for PEG separation. **a** Schematic for sorting polymers with different terminal groups using a rigid MOF. **b** 2-D $^1$H–$^{13}$C HETCOR NMR spectrum of **1a** treated with H–PEG–H. **c** DSC heating curves of neat PEG (dotted line) and **1a** after the heating treatment with PEG (solid line). H–PEG–H, Tr–PEG–Tr, mixture of H–PEG–H and Tr–PEG–Tr, and **1a** with H–PEG–H and Tr–PEG–Tr after washing. $M_n$ of PEG used in these experiments was 2 kDa. **d**−**f** MD simulation snapshots of **1a** with **d** H–PEG–H and **e**, **f** Tr–PEG–Tr at 373 K. MD structure of Tr–PEG-Tr near the surface (**f**) showed that selective exclusion of Tr–PEG-Tr was attributed to the steric hindrance of terminal groups as well as the π-π interaction between Tr groups and the ligands

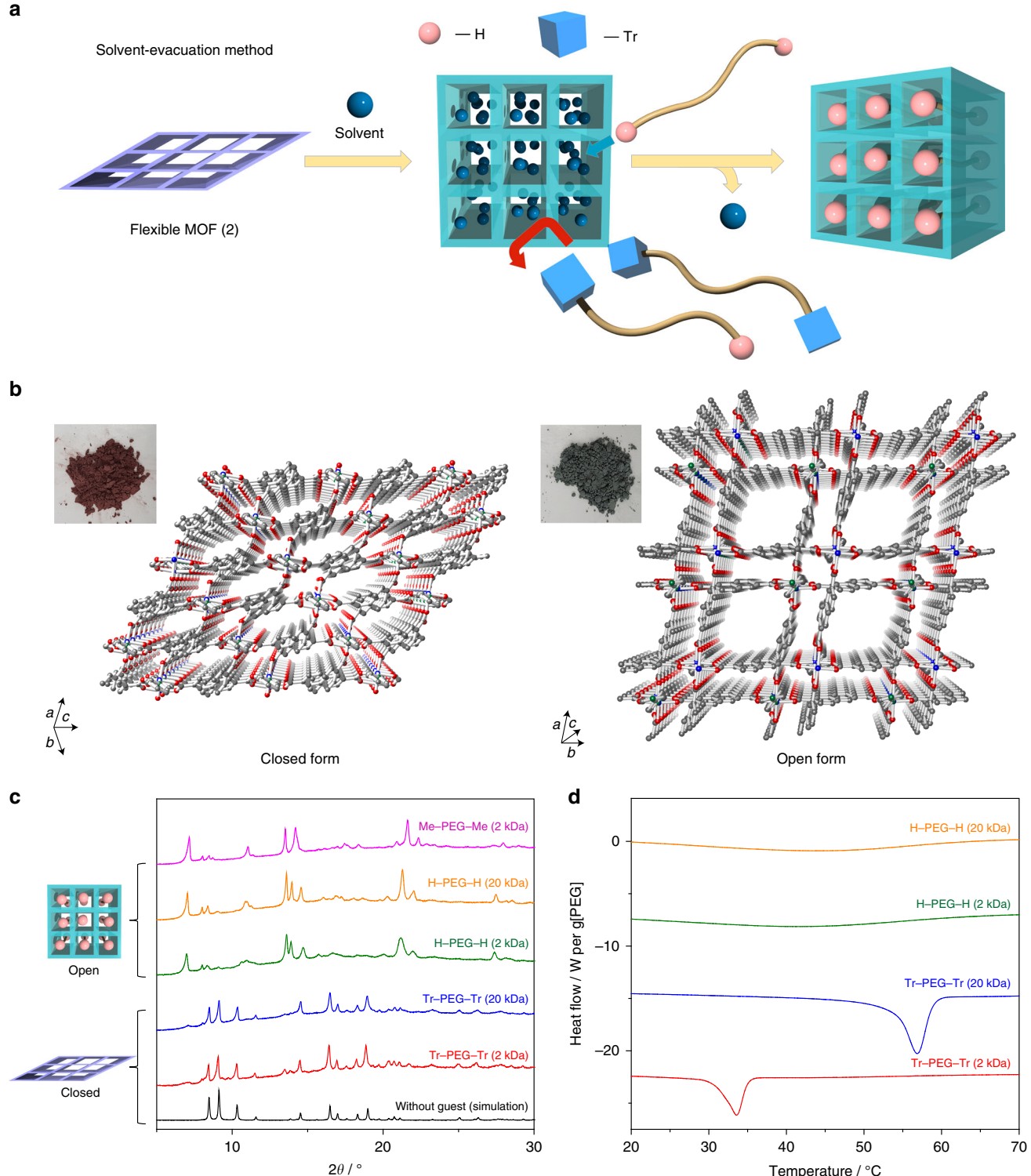

**Fig. 2** Introduction of PEG into **2** using solvent-evacuation method. **a** Schematic for sorting polymers with different terminal groups using a flexible MOF system (solvent-evacuation method). **b** Crystal structure and photograph of **2** (left: closed form, right: open form). **c** PXRD patterns of dried **2**, and **2** with PEG obtained by the solvent-evacuation method. **d** DSC profiles of **2** and PEG composites. The traces correspond to the first cycle of the analysis, and samples were not thermally annealed before the measurement

, showed a markedly different insertion behavior. Following the same protocol, we prepared a composite between **1a** and Tr–PEG–Tr. SEM and particle size analysis indicated the agglomeration of MOF particles, possibly stuck together by Tr–PEG–Tr outside the MOF (Supplementary Figs 3, 4). No correlation signal of Tr–PEG–Tr and **1a** was detected in the 2-D

[1]H–[13]C HETCOR NMR spectrum (Supplementary Fig. 10). Furthermore, the endothermic peak of neat Tr–PEG–Tr was still observed in the DSC profile of the composite, which remained unchanged even after thermal annealing (Fig. 1c and Supplementary Fig. 6). Thus, functionalization by Tr groups resulted in the exclusion of the PEG chains from **1a**. Since the PEG chains must

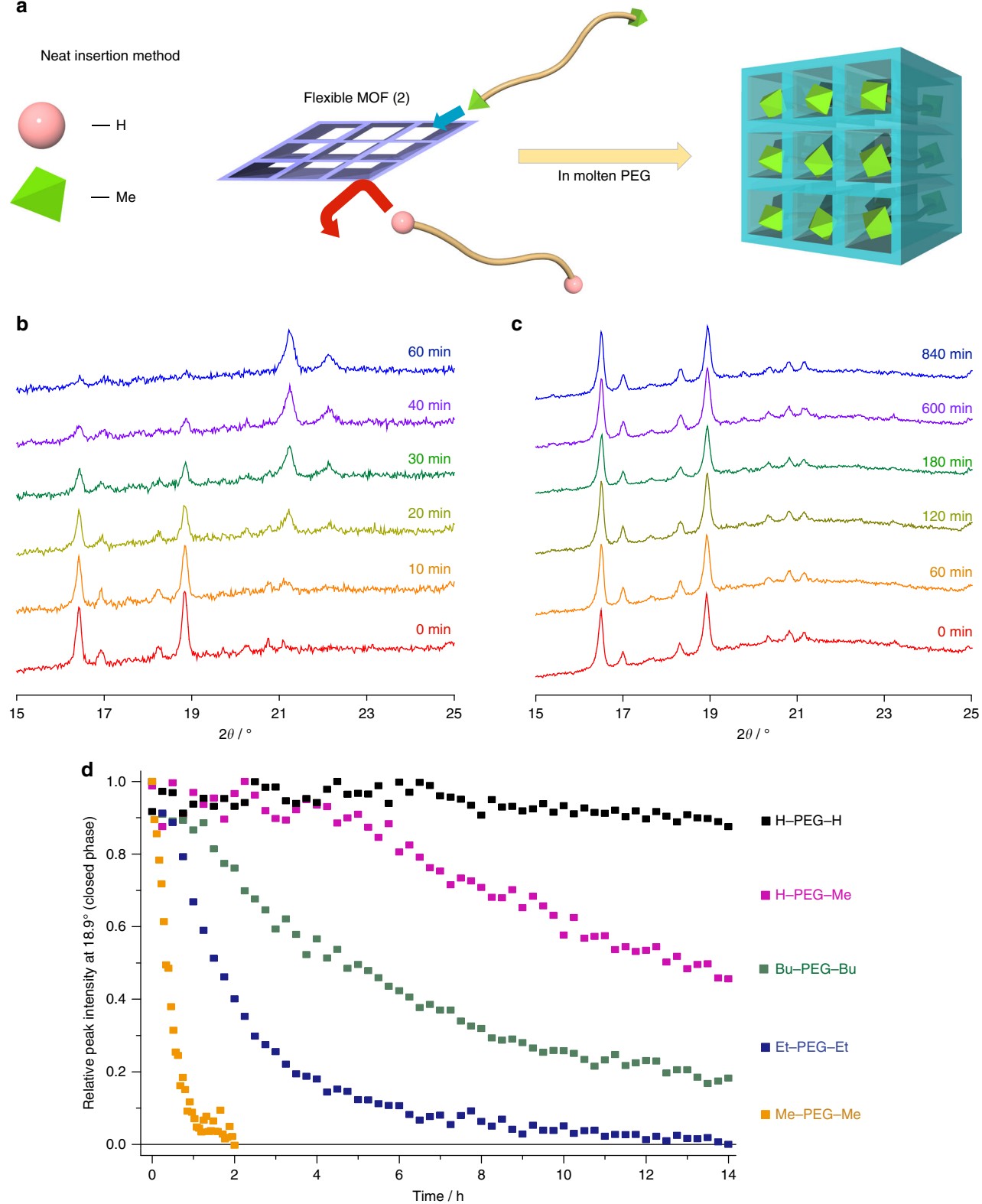

**Fig. 3** Introduction of PEG into **2** using neat insertion method. **a** Schematic for sorting polymers with different terminal groups of similar sizes using a flexible MOF system (neat insertion method). **b**, **c** Time-resolved in situ PXRD patterns of **2** in presence of Me–PEG–Me and H–PEG–H, respectively. **d** Evolution of the proportion of closed-phase **2** (determined by following the intensity of the characteristic peak at 18.9°) in contact with several PEGs at 45 °C. $M_n$ of PEG used in these experiments was 1 kDa

pass through the nanowindows of **1a** from their extremities, rejection of Tr–PEG–Tr was likely caused by the steric hindrance of the bulky terminal group. This was supported by the observation that the structural analog [Zn$_2$(1,4-benzenedicarboxylate)$_2$triethylenediamine]$_n$ (**1b**, Supplementary Fig. 1) with larger nanopores (pore size = 7.5 × 7.5 Å$^2$)[37] enabled the encapsulation of Tr–PEG–Tr, as a result of a pore size slightly larger than the Tr group (Supplementary Figs 2–5).

To provide an insight into how the MOF channels can discriminate the terminal groups of PEGs, we investigated the insertion and diffusion behaviors of the PEG chains using a molecular dynamics (MD) simulation[36]. H–PEG–H on the [001] surface of **1a** spontaneously permeated and diffused into the nanochannels, which was in stark contrast to the case of Tr–PEG–Tr (Fig. 1d). The MD structure revealed that selective exclusion of Tr–PEG–Tr was attributed to the steric hindrance of terminal groups as well as the π–π interaction between Tr groups and the ligands (Supplementary Fig. 11). In contrast, we could observe the spontaneous permeation of Tr–PEG–Tr into **1b**, which is consistent with the experimental results (Supplementary Fig. 11).

This method was also applied to the recognition of a hemifunctional PEG, Me–PEG–Tr, terminated with a single Tr group. The large Tr group cannot access the nanochannels of **1a**, and prevents the inclusion of whole polymer chains in the MOF, as revealed by DSC and 2-D NMR (Supplementary Figs 9 and 10). This was also confirmed by MD simulation, in which the Me-terminus of PEG entered the channels; however, the adsorption of Me–PEG–Tr remained limited to the surface of MOF crystals only, as the Tr-terminus remained excluded by the pore windows (Supplementary Fig. 11). Remarkably, **1a** also recognized the various terminal groups of PEG with much higher molecular weight. This is a significant advantage to technologically relevant separation because the concentration of terminal groups decreases with increasing the molecular weight of polymers. Therefore, such polymers have quite similar properties, despite the different terminal groups, which makes it extremely difficult to recognize the dissimilarity using conventional methods (Supplementary Fig. 12). For instance, the mixture of H–PEG–H and Tr–PEG–Tr exhibited a single melting peak, showing miscibility of each polymer at the nanometer level (Fig. 1c). However, it is notable that **1a** allowed for the full encapsulation of H–PEG–H with a molecular weight of 20 kDa in the pores, while Tr–PEG–Tr with the same molecular weight was rejected completely by **1a** (Supplementary Figs 5, 9, and 10).

The Tr group is commonly used for the protection of hydroxyl groups in functional molecules and macromolecules[38], which often requires rigorous purification of the reaction mixtures for further applications. Selective inclusion of PEG into **1a** allowed for the successful separation of pristine and Tr-modified PEGs from their mixtures. In this protocol, **1a** was treated with a mixture of H–PEG–H and Tr–PEG–Tr, followed by thermal annealing. After incorporation, the DSC scan for the resulting composite exhibited a single melting peak at the same position as that of pure Tr–PEG–Tr, which highly contrasted to the PEG mixture in the initial bulk state (Fig. 1c). This was the clear indication of selective exclusion of Tr–PEG–Tr from the nanochannels of **1a**. Furthermore, Tr–PEG–Tr outside the channels was fully collected by washing the composites with $CH_2Cl_2$, as evidenced by the disappearance of the PEG melting peak in the DSC profile (Fig. 1c) and by $^1$H-NMR (Supplementary Fig. 13). Purified H–PEG–H was quantitatively collected from **1a** after digestion of the washed composites in a 0.05 M tetrasodium ethylenediaminetetraacetate ($Na_4$EDTA) solution. It should be noted that this system efficiently sorted even PEG with a high molecular weight (20 kDa), providing analytically pure Tr–PEG–Tr from the mixture with H–PEG–H (Supplementary Fig. 13). These results demonstrate the high potential of rigid MOFs to realize simple, yet very efficient procedures of purification for polymers with termini having different steric hindrances.

**Sorting of PEG by a flexible MOF**. The selective recognition and separation of PEG was also performed using a flexible MOF, [$Co_2$(2,6-naphthalenedicarboxylate)$_2$(4,4′-bipyridine)]$_n$ (**2**, Fig. 2a, b)[39,40]. Without guest molecules, **2** presented a collapsed form with very narrow pores. Similarly to **1a**, Me–PEG–Me and H–PEG–H dissolved in MeCN were successfully introduced into **2** by removing the solvent under vacuum. During this process, powder X-ray diffraction (PXRD) was a valuable method to assess the accommodation of PEG, as the closed **2** was converted into another phase with a different diffraction pattern (Fig. 2c). The phase transition also caused a dramatic color change of the MOF, from purple to green. Structural determination was performed using single crystals of **2** including H–PEG–H (0.6 kDa) (Supplementary Fig. 14 and Supplementary Table 1). The simulated PXRD pattern was in agreement with experimental data collected with longer PEGs, indicating that the structure of **2** including PEG did not depend on the polymer length. PEG inclusion resulted in the opening of pores along the c-axis (i.e., parallel to the bipyridine ligands), with a size of $4.7 \times 4.7$ Å$^2$. Other channels along the [101] and [011] directions, with smaller diameters, were also present and occupied by PEG electron density (Supplementary Fig. 15). This means that **2** was effectively 3D porous for PEG.

The maximal PEG content of **2** was $30 \pm 2$ wt%, as determined by DSC (Supplementary Fig. 16). As distinct from **1a**, the introduction of PEG into **2** did not require thermal annealing. Surprisingly, H–PEG–H with a high molecular weight (20 kDa) could be introduced quantitatively at the maximal loading by evacuation of MeCN at room temperature, 30 °C below the melting point of the PEG in the bulk state (Fig. 2c, d). The introduction of H–PEG–H by solvent evacuation was thus highly efficient for this MOF. Owing to the structural flexibility of **2**, the diffraction patterns of the closed and open phases could be monitored in situ, providing more insight into the polymer insertion (Supplementary Fig. 17). When a solvent was present, **2** was opened by solvent molecules and no PEG inclusion was detected. Introduction of PEG into **2** occurred within minutes under vacuum, indicating that the phenomenon was actively driven by solvent evacuation. Interestingly, removal of solvent from **2** was significantly slower in the absence of PEG, suggesting that PEG chains contributed to the solvent displacement and promoted their own insertion. When **2** was treated with Tr–PEG–Tr in the same conditions, the polymer remained excluded from the channels of the MOF. This was noticeable by the sole presence of closed-form **2** after evacuation of the solvent (Fig. 2c), and the endothermic peaks corresponding to melting of the bulk PEG (Fig. 2d). As with the rigid **1a**, this exclusion was likely the result of a steric hindrance of the bulky Tr groups that prevented threading through the nanochannels of open **2**.

Because of the interesting loading mechanisms, H–PEG–H chains were efficiently introduced into MOF pores, permanently opened in **1a** and preopened with solvents in **2**. These systems proved very promising for the separation of functionalized PEG based on steric hindrance, as bulky terminal groups were rejected by pore windows. However, using this method, PEG with terminal groups of similar sizes, but different chemical properties, cannot be separated. Nevertheless, the separation of small molecules with similar size is one of the most attractive applications of MOFs. For this precision separation, flexible MOFs are especially advantageous, as they represent dynamic structural changes induced by guest molecules (so-called breathing or gate-opening effects)[41,42]. Such flexible structures benefit from an adaptive porosity that can be opened only by guests with adequate natures. This is notably the case for **2** that can absorb readily solvents of intermediate polarity (e.g., MeCN, DMF, $CH_2Cl_2$, EtOH) to become the open phase, but remains closed for highly polar (e.g., water, ethylene glycol) and nonpolar (e.g., hexane) solvents (Supplementary Fig. 18). This advantage was, however, neglected when using the solvent-evacuation method because **2** was preopened by solvents. Based on these considerations, we realized the unprecedented separation of PEG with only

a marginal difference at the terminal groups. The flexible **2** successfully allowed for the critical discrimination of PEG modified with different alkyl groups, such as H, Me, Et, and *n*-Bu. In this system, **2** with closed pores was treated with neat molten PEG, since the presence of solvents would have caused pore opening (Fig. 3a). Again, the inclusion of PEG into **2** was monitored by in situ PXRD measurement. Figures 3b, c show the evolution of the PXRD patterns of **2** in the presence of Me−PEG−Me and H−PEG−H, respectively. Time-resolved PXRD patterns indicated the progressive conversion of **2** from the closed to the open phases upon treatment with Me−PEG−Me, following an apparent first-order reaction kinetics (Fig. 3d). Progressive introduction of Me−PEG−Me was also confirmed by DSC, and proceeded until a maximal loading of 29.6 ± 0.8 wt% was reached (Supplementary Fig. 19). In the absence of solvents, the introduction of PEG proceeded only following the passive diffusion mechanism, and thus the insertion speed increased sharply when decreasing the molecular weight of Me−PEG−Me, occurring almost instantly for $MeO(CH_2CH_2O)_4Me$. In contrast, H−PEG−H was never included into **2**, even for $HO(CH_2CH_2O)_4H$ with a very low molecular weight (Supplementary Fig. 18). Because initial **2** presented no pore suitable for PEG, the framework of **2** needed to be opened by adequate chemical groups before the PEG insertion could proceed. Following a trend similar to organic solvents, the terminal OMe group with intermediate polarity presented a higher affinity for **2**, while highly polar OH group at the PEG terminal could not open the pores for accommodation. Here we also examined Et−PEG−Et and Bu−PEG−Bu for the introduction into **2**. In all cases, the terminal moieties have the same thickness of one methylene unit, not thicker than the main chain. Thus, recognition of PEG with different terminal alkyl groups does not occur through steric hindrance; however, the dynamic **2** has the specific preference for the accommodation of guests with different terminal polarities. For a same chain length, the speed of PEG insertion was found to be highly dependent on the nature of the terminal alkyl groups, where the adsorption speed of PEG decreased in the following order: Me>Et>Bu>>H (Fig. 3d and Supplementary Fig. 20). In addition to homofunctional PEG, introduction experiments using a heterofunctional PEG were performed. Interestingly, H−PEG−Me could be encapsulated in **2**, but its inclusion was significantly slower than Me−PEG−Me, possibly suggesting the unidirectional insertion from the Me-terminus (Fig. 3d and Supplementary Fig. 20). Given enough time, all the alkylated PEGs were quantitatively introduced in **2** to reach almost full loadings. Note that H−PEG−H and the alkylated PEGs could all be easily and readily included in **2** with the solvent-evacuation method (Supplementary Fig. 21). Formation of **2** and PEG composites was thus thermodynamically favored, showing that the critical selectivity for alkyl termini in neat system was attributable to kinetic factors.

Owing to the clear difference between their insertion speeds, we accomplished the kinetic separation of Me−PEG−Me from its mixture with H−PEG−H (Supplementary Table 2 and Supplementary Fig. 22). After insertion, the PEG mixture inside **2** was found to be significantly enriched in Me–PEG–Me, while a depletion was observed outside. The calculated selectivity for the insertion of Me–PEG–Me into **2** was at least >5.2. This high selectivity upon a single separation step for a very minute difference of composition between the two PEGs prefigures the high potential of flexible MOFs for the separation of macromolecules by chemical nature, as a complement to the separation by pore size exclusion.

## Discussion

Beyond the use of MOFs for the separation of gases and small molecules, we demonstrated their high potential for the separation of polymers differing only by their terminal groups. In this system, insertion of polymer chains into MOFs proceeds by its extremities, allowing for the separation of functionalized PEG with different terminal groups of various natures and sizes. It was noteworthy that MOFs with rigid pores could critically discriminate the terminal groups of PEG based on steric hindrance. In addition, we could advantageously utilize the dynamic character of flexible MOFs for the precise recognition of terminal groups with a similar size, but slightly different polarity. These two separation modes, pivotal for the separation of small molecules by MOFs, maintain here their capability to discriminate terminal groups even if they are attached to long polymer chains. Use of MOFs may thus be at the core of a highly desired method for the robust and systematic separation of polymers, and notably for PEG, whose ubiquitous use in biology brings the need for highly pure materials. Thanks to the diversity of MOFs, our strategy can be further extended to a wide variety of polymer separations, opening exciting perspectives for highly applicable purification systems of polymers.

## Methods

**Materials.** $Zn(NO_3)_2 \cdot 6H_2O$, $Co(NO_3)_2 \cdot 6H_2O$, and triethylenediamine were purchased from Wako Pure Chemicals Industries. 1,4-naphthalenedicarboxylic acid, 2,6-naphthalenedicarboxylic acid, 4,4′-bipyridine, trimethylamine, iodomethane, bromoethane, bromopropane, and trityl chloride were purchased from Tokyo Chemical Industries. H–PEG–H and Me–PEG–Me were purchased from Merck. H–PEG–Me was purchased from Creative PEGworks. All reagents were used as received without purification. MOFs were prepared according to literatures[34,37,40].

**Insertion of PEG by solvent-evacuation method.** In a typical insertion, a known amount (ca. 100 mg) of activated (i.e. guest-free) MOF was suspended in anhydrous MeCN. The desired amount of PEG dissolved in anhydrous MeCN was then added, and the solvent was evacuated progressively under reduced pressure (0.3 kPa). In the case of **1a** and **1b**, a thermal annealing (70 °C for 12 h under reduced pressure) was accomplished after solvent evacuation to drive the complete insertion of PEG. In the case of **2**, the insertion could proceed quantitatively even at room temperature, so the thermal annealing step was not necessary. Note that in this case, **2** was spontaneously opened by solvent before the insertion proceeded.

**Direct insertion of neat PEG.** Insertion was realized by maintaining a mixture of activated MOF and PEG above its melting temperature. Note that to guarantee an efficient wetting of the MOF surface by molten PEG, the polymer was added in excess compared with the maximal capacity of the MOF.

**Separation of PEG using 1a.** Degassed **1a** (100 mg) was immersed in a MeCN solution of PEG (2 kDa) mixture, H–PEG–H (10 mg) and Tr–PEG–Tr (10 mg). MeCN was removed by evacuation at 70 °C for 12 h, allowing for selective incorporation of H–PEG–H into the nanochannels of **1a**. To remove Tr–PEG–Tr outside **1a**, the resulting composite was suspended in $CH_2Cl_2$ (10 ml), filtrated, and dried under reduced pressure, affording **1a** containing H–PEG–H. Digesting the washed composite in $Na_4EDTA$ solution allowed for the quantitative recovery of H–PEG–H.

## Data availability

The X-ray crystallographic coordinates for the structure of **2** with PEG have been deposited at the Cambridge Crystallographic Data Center, under deposition number 1836242. These data can be obtained free of charge from The Cambridge Crystallographic Data Centre via www.ccdc.cam.ac.uk/data_request/cif. All relevant data are available from the authors upon reasonable requests.

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

## Acknowledgements

This work was supported by the JST, CREST program (JPMJCR1321), and a Grant-in-Aid for Science Research on Innovative Area "Coordination Asymmetry" (JP16H06517) from the Ministry of Education, Culture, Sports, Science and Technology, Government of Japan.

## Author contributions

T.U. conceived and directed the project. B.L. designed and performed the experiments. C. W. and T.K. carried out the experiments using rigid MOFs. S.M. contributed to single-crystal X-ray diffraction analysis. M.T. and M.N. performed the computational analysis. B.L, T.K., and T.U. discussed results and wrote the paper.

## Additional information

**Competing interests:** The authors declare no competing interests.

