## [Peer Review File · Nature Communications]

Reviewers' Comments:

Reviewer #1:

Remarks to the Author:

The manuscript reports the efficient separation of high molecular weight polymers with different terminal groups using MOFs. The authors performed extensive experiments and MD simulation on the separation mechanism of PEG, and characterized the materials with a range of techniques. In addition, the authors performed detailed analysis of DSC trace, NMR and in-situ PXRD. The protocol for the efficient high molecular weight polymers using MOFs is indeed remarkable. The paper is significant and certainly of general interest to the field and suitable for Nature Communication.

However, the feasibility of the separation experimental needs to be emphasized. The authors make a claims about efficient polymer separation based on the single compound experiments. In the mixture, however, the separation efficiency may not good as the authors claimed since fouling by larger terminal group may occur. Therefore, I would recommend the authors to provide a mixture separation experiments (e.g. breakthrough curve) in order to meet the high standard of Nature Communications.

Minor comments:

In supplementary figure 5, Tr-PEG-Tr (2 kDa) (green) should be '20 kDa'

Reviewer #2:

Remarks to the Author:

In their paper the authors describe experiments where different types of MOF powders were exposed to linear polymers of different lengths and different end groups. It was found that the polymers diffuse into the MOF materials, in case of flexible MOFs this loading caused a change in structure which was detected by X-ray diffraction. The authors could demonstrate that the time constant describing the loading is strongly different for different polymers; even for small modifications of the polymer, e.g. variation of the terminating group, substantial acceleration or delay of indiffusion was observed. The authors propose that these different time constants upon loading may be used for separation of very similar polymers. Although the method is certainly limited to linear polymers with rather small diameter, for a large class of polymers including PEG this is a very interesting approach. The experiments have been carried out with great detail and were described very carefully.

I believe this paper is well suited for publication in Nature Communications.

Reviewer #3:

Remarks to the Author:

The manuscript "Selective Sorting of Polymers with Different Terminal Groups using Metal-Organic Frameworks" by Uemura and coworkers addresses the size specific incorporation of PEG chains into MOFs. Overall, the concept and results are a significant contribution to the field and certainly suited for Nat. Commun. Nevertheless, there are some comments that the authors should address/consider before publication.

General comments:

1. Can the authors comment on the mechanism of chain inclusion? Is it some kind of reptation movement?
2. The thermodynamics of insertion should be elaborated or discussed further. Why does polymer move inside of the channel at all?
3. The authors should have a look at the following papers that deal with polymer-MOF interactions:
<https://pubs.acs.org/doi/abs/10.1021/jacs.7b12633>
<https://onlinelibrary.wiley.com/doi/abs/10.1002/anie.201607329>
<https://onlinelibrary.wiley.com/doi/abs/10.1002/adfm.201605465>
4. Does MOF particle size have an influence? As the polymers enter the MOF from the outer surface there should be an effect of outer MOF surface. I think that point should be discussed further. For example how many pores are accessible to the PEG chains regarding the outer surface of the MOF.
5. How was the purity of the PEG materials assessed?
6. Is there any other way to investigate the efficiency of polymer loading? Maybe with TGA?

Specific comments:

7. In the introduction the authors mention "a different nature" the expression seems a bit out of place to me

Reviewers' comments:

Reviewer #1 (Remarks to the Author):

The manuscript reports the efficient separation of high molecular weight polymers with different terminal groups using MOFs. The authors performed extensive experiments and MD simulation on the separation mechanism of PEG, and characterized the materials with a range of techniques. In addition, the authors performed detailed analysis of DSC trace, NMR and in-situ PXRD. The protocol for the efficient high molecular weight polymers using MOFs is indeed remarkable. The paper is significant and certainly of general interest to the field and suitable for Nature Communication.

We thank the reviewer for this comment.

However, the feasibility of the separation experimental needs to be emphasized. The authors make a claims about efficient polymer separation based on the single compound experiments. In the mixture, however, the separation efficiency may not good as the authors claimed since fouling by larger terminal group may occur. Therefore, I would recommend the authors to provide a mixture separation experiments (e.g. breakthrough curve) in order to meet the high standard of Nature Communications.

We agree with the reviewer's comment on the importance of showing actual separation of PEG mixtures. Indeed, the initial manuscript described the separation experiments using rigid and flexible MOFs in closed batch system. In these experiments, MOFs were treated with PEG mixtures followed by thermal annealing for selective insertion of PEG, resulting in the production of highly purified PEG. The efficiency of the PEG purification was assessed by DSC and NMR analyses (Figure 1C, Supplementary Figure S13, S22). The composition of PEG mixtures could be determined using NMR spectroscopy, in which the peak integral of terminal groups was compared to that of main chain.

Separation of PEG on the basis of terminal size difference was very efficient using a rigid MOF (**1a**), so that analytically pure Tr-PEG-Tr and H-PEG-H were collected outside and inside the MOF, respectively. In our experimental conditions (overnight heating at 100 °C), surface fouling by non-inserted PEG did not hinder the insertion of H-PEG-H, showing the potential of very efficient procedure for polymer separation. In the case of a flexible MOF (**2**), the kinetic separation of mixtures of Me-PEG-Me and H-PEG-H was achieved owing to the large difference in their insertion speed. Inclusion of Me-PEG-Me was much faster than H-PEG-H despite a very minute difference between their terminuses. We could thus observe an efficient separation of mixtures favouring Me-PEG-Me insertion with a high selectivity of 5.2.

Dynamic separation experiments (e.g. breakthrough) are challenging for pure polymers because their high viscosity (or even their solid state) prevents an efficient flow. Use of eluting solvents would enable the separation of polymers in chromatography experiments, which is currently in development and will be the object of future articles.

In the revised manuscript, the paragraphs about separation have been rewritten to put more emphasis on the separation procedures that were performed (Page 4, line 21-32; Page 6, line 21-28).

Minor comments:

In supplementary figure 5, Tr-PEG-Tr (2 kDa) (green) should be '20 kDa'

The measurements on Supplementary Figure 5B was performed with the same PEG chain length (2 kDa). We demonstrate that PEGs with both H and Tr terminal groups can be inserted in the MOF **1b** with larger pores.

Reviewer #2 (Remarks to the Author):

In their paper the authors describe experiments where different types of MOF powders were exposed to linear polymers of different lengths and different end groups. It was found that the polymers diffuse into the MOF materials, in case of flexible MOFs this loading caused a change in structure which was detected by X-ray diffraction. The authors could demonstrate that the time constant describing the loading is strongly different for different polymers; even for small modifications of the polymer, e.g. variation of the terminating group, substantial acceleration or delay of indiffusion was observed. The authors propose that these different time constants upon loading may be used for separation of very similar polymers. Although the method is certainly limited to linear polymers with rather small diameter, for a large class of polymers including PEG this is a very interesting approach. The experiments have been carried out with great detail and were described very carefully. I believe this paper is well suited for publication in Nature Communications.

We thank the reviewer for this comment. Now that the fundamental principles have been highlighted, translation of the technique to other polymers, to investigate the broadness of its scope, is indeed a priority.

Reviewer #3 (Remarks to the Author):

The manuscript "Selective Sorting of Polymers with Different Terminal Groups using Metal-Organic Frameworks" by Uemura and coworkers addresses the size specific incorporation of PEG chains into MOFs. Overall, the concept and results are a significant contribution to the field and certainly suited for Nat. Commun. Nevertheless, there are some comments that the authors should address/consider before publication.

We thank the reviewer for this comment.

General comments:

1. Can the authors comment on the mechanism of chain inclusion? Is it some kind of reptation movement?

The mechanism of inclusion is indeed a reptation of the chain through multiple MOF windows. This is a key aspect of the separation of PEG, as the chains have to “crawl” through a series of MOF nanowindows, each one constituting a barrier for Tr groups. The mechanistic study by MD simulation of PEG insertion inside a MOF have already been reported (Uemura *et al. J. Phys. Chem. C* **2015**, *119*, 21504). The simulation revealed that H-PEG-H chains permeate and diffuse into the nanochannels of MOF, which was also observed in our current study (Figure 1D).

2. The thermodynamics of insertion should be elaborated or discussed further. Why does polymer move inside of the channel at all?

Thanks to the valuable comment from this reviewer, we performed calorimetry experiments using DSC. Guest-free **1a** and solid PEG were placed together in a DSC crucible. A very slow heating ramp (1 °C min⁻¹) was used to allow for the quantitative insertion of PEG during the first DSC cycle. Upon heating, the sample exhibited an endothermic peak corresponding to PEG melting, followed by an exothermic peak originating from PEG insertion. The integral of the exothermic peak corresponds the heat released upon introduction of PEG. For the introduction of H-PEG-H in **1a**, the heat of adsorption was determined as 175 J per g of PEG inserted, corresponding to 7.7 kJ per mol of the repeating unit in PEG. This value is comparable to that obtained from MD simulation for adsorption of oligo(ethylene glycol) (14mer) in a MOF (Uemura *et al. J. Phys. Chem. C* **2015**, *119*, 21504).

We added these descriptions in the revised manuscript (Page 3, lines 20-22; Supplementary Figure 7). In this revision, we also modified Supplementary Figure 8 to clarify the difference between thermodynamics and kinetics studies using DSC.

3. The authors should have a look at the following papers that deal with polymer-MOF interactions:
<https://pubs.acs.org/doi/abs/10.1021/jacs.7b12633>
<https://onlinelibrary.wiley.com/doi/abs/10.1002/anie.201607329>
<https://onlinelibrary.wiley.com/doi/abs/10.1002/adfm.201605465>

We are grateful to the reviewer for these indications. These articles described very interesting achievements, and presented diverse approaches of the preparation of Polymer@MOF composites (encapsulation of micelles in a MOF, synthesis of a MOF in micelles, and conversion of polymer/metal hydroxide nanocomposite). They have thus been cited in the revised manuscript (Ref. 23, 25, 26).

4. Does MOF particle size have an influence? As the polymers enter the MOF from the outer surface there should be an effect of outer MOF surface. I think that point should be discussed further. For example how many pores are accessible to the PEG chains regarding the outer surface of the MOF.

We thank the reviewer for this suggestion.

The MOF particles used in this work had a diameter of a few micrometers (ca. 5-50 μm). This is typical for MOF syntheses that are not designed to obtain nanoparticles. As such, the outer surface area of the MOF

particles is very low (below 1 m²/g). Changing the particles size in the micrometer range (1 to 100 um) would probably have only a limited effect.

The use of MOF nanoparticles with a high outer surface area would indeed be very interesting. Smaller particle size would bring a higher accessibility of guests, and possibly new effects as surface adsorption would become non-negligible. However, preparing the nanosized MOFs requires new synthetic pathways, and not yet be reported for the MOFs used in this work. This would also bring uncertainty about the exact phase considered (density of defects mainly). In addition, flexible MOFs might change their adsorption behaviour when nanosized (Sakata *et al. Science* **2013**, 339, 193), making the study using MOF 2 difficult to transpose. The impact of particle size will thus be studied in a future work.

5. How was the purity of the PEG materials assessed?

The purity of functionalized PEG was assessed by NMR spectroscopy. The conversion yield was determined by comparing the NMR integral of the terminal groups to that of the main chain, showing that all the PEGs used here had quantitative conversions. We added these descriptions in Supplementary Information (1. Methods).

6. Is there any other way to investigate the efficiency of polymer loading? Maybe with TGA?

Although TGA was naturally envisaged to evaluate guest loading, the decomposition temperature of PEG was close to those of MOFs, precluding the determination of the exact amount of PEG loading. Thus, we did not include TGA in this manuscript, but used more efficient characterization techniques.

To assess the effective PEG insertion, quantitative techniques sensitive to the existence of PEG inside and outside the MOFs were performed using XRD (Figure 3B, 3C) and DSC (Figure 1C, 2D), respectively. Multinuclear 2D NMR was very powerful to prove the efficient loading of PEG chains inside the MOFs (Figure 1B). After etching the MOF-PEG composites in DMSO-d₆/DCI mixture, we carried out ¹H NMR spectroscopy to compare the integrals of PEG and MOF ligands, which enabled the precise determination of the total loading amount of PEG (Supplementary Figure S13, S22).

Specific comments:

7. In the introduction the authors mention "a different nature" the expression seems a bit out of place to me

According to the reviewer's comment, we now refer to terminal groups with "a different polarity" (Page 2, line 41).

Reviewers' Comments:

Reviewer #1:

Remarks to the Author:

Required revision is made. Now the paper can be accepted for publication.

Reviewer #3:

Remarks to the Author:

The revised version of the manuscript "Selective Sorting of Polymers with Different Terminal Groups using Metal-Organic Frameworks" by Uemura and coworkers addresses all relevant issues. I recommend publication of the manuscript in its current version.